# The Role of Interoceptive Sensitivity and Hypnotizability in Motor Imagery

**DOI:** 10.3390/brainsci14080832

**Published:** 2024-08-19

**Authors:** Eleonora Malloggi, Žan Zelič, Enrica Laura Santarcangelo

**Affiliations:** 1Department of Physics, University of Trento, 38122 Trento, Italy; eleonora.malloggi@unitn.it (E.M.); zan.zelic@unitn.it (Ž.Z.); 2Department of Translational Research and New Technologies in Medicine and Surgery, University of Pisa, Via San Zeno 31, 56127 Pisa, Italy

**Keywords:** hypnotizability, motor imagery, interoceptive sensitivity, EEG power spectral density, MAIA

## Abstract

Motor imagery (MI) requires the mental representation of the body, obtained by integrating exteroceptive and interoceptive information. This study aimed to investigate the role of interoceptive sensitivity (IS) in MI performed through visual and kinesthetic modalities by participants with low (lows, *N* = 26; SHSS: A, *M* + *SD*: 1.00 + 1.52), medium (mediums, *N* = 11; SHSS: A, 6.00 + 0.77) and high hypnotizability scores (highs, *N* = 16; SHSS:A, 9.75 + 1.24), as measured by the Stanford Hypnotic Susceptibility Scale: Form A. The three groups displayed different MI abilities and IS levels. The efficacy of MI was measured using the chronometric index and self-reported experience, while IS was measured using the Multidimensional Assessment of Interoceptive Awareness (MAIA) questionnaire. Alpha and beta power spectrum densities (PSDs) were extracted from the EEG signals acquired during baseline, actual movement and visually and kinesthetically imagined movements. The chronometric indices do not reveal significant differences between groups and imagery modalities. The self-report MI efficacy indicates better kinesthetic imagery in highs and mediums than in lows, and no modality difference among lows. The MAIA dimensions sustain the differences in subjective experience and almost all the EEG differences. The latter are slightly different in highs, mediums and lows. This is the first report of the major role played by IS in MI and strongly supports the theory of embodied cognition.

## 1. Introduction

Motor imagery (MI) is the mental simulation of a motor action without its actual execution [1]. It is used in neuro-rehabilitation therapies, including brain–computer interface interventions [2]. However, its efficacy is difficult to predict due to the large variability of motor imagery abilities in the general population [3,4]. 

Actual and imagined movements have been associated with different electroencephalographically recorded (EEG) patterns during their preparatory, execution and post-execution phases. Generally, alpha and beta desynchronization is observed during movement preparation and execution phases, whereas beta rebound occurs at the end of the movement [5,6]. Similar findings are reported for MI, with a decrease in EEG power (desynchronization), indicating increased cortical activity [7,8] and an increase representing the maintenance or inhibition of cortical activity [7,9,10]. 

According to the functional equivalence theory [11], the more similar the cortical activity during actual and imagined movement, the more efficacious the MI [12]. The difference between the duration of actual and imagined movements can also be used as a behavioral measure of the efficacy of MI (chronometric index). Indeed, actual and well-imagined actions have been shown to share time duration and autonomic responses in athletes [13] and in neurological patients [14,15]. In addition, the intensity of the experience of MI can be assessed by self-reports. Nonetheless, the correspondence between behavioral, EEG and subjective measures of MI is still debatable. It has been suggested that MI should be considered from multiple perspectives, including self-reports, chronometric indices, EEG activities and autonomic responses [16]. 

### 1.1. MI and Interoceptive Sensitivity

Efficacious MI requires a correct body schema, influenced by interoception [17], which is described as the »sense of the body« [18]. Interoception includes the dimensions of accuracy (IA, measured by the heartbeat counting task and heartbeat-evoked cortical potential (HEP)), sensitivity (IS, measured by self-report questionnaires) and awareness (the correspondence between IA and IS) [19]. IS is generally measured by the Multidimensional Assessment of Interoceptive Awareness (MAIA, [20]), which consists of eight scales (noticing, not distracting, not worrying, attention regulation, emotional awareness, self-regulation, body listening, trusting), indicating the mode of the individual interpretation of interoceptive signals. 

The insular cortex is the brain structure most involved in interoception. It receives all sensory inputs, including somesthetic information, is involved in the sensory context relevant to voluntary movements [21] and participates in decision-making, social and risky behavior [22]. The activity of the anterior insula is involved in interoceptive accuracy [23], whereas its functional connectivity with the cerebellum, ventral striatum, brainstem and prefrontal cortex has been reported to be positively correlated with interoceptive sensitivity [24]. 

The body sensations experienced during motor imagery with different emotional content can increase or decrease the functional connectivity between the insula and the dorsomedial frontal cortex [22]. Additionally, an impaired body schema has been observed in patients with anorexia nervosa [25], who are also less accurate than control subjects in MI and less successful in the mental rotation of human figures [26]. Nonetheless, no study investigated the role of interoceptive sensitivity in motor imagery, despite the well-recognized role of the insula in this dimension of interoception.

### 1.2. Hypnotizability, MI and IS

Hypnotizability is a psychophysiological trait stable throughout life [27] and best-known for the ability of highly hypnotizable individuals to control pain through cognitive strategies [28]. It is measured by standardized scales classifying high (highs), medium (mediums) and low hypnotizable individuals (lows). Different hypnotizablity levels are associated with several behavioral and brain morpho-functional characteristics [29]. Among the former, there are differences in sensorimotor integration, vascular control and sensitivity of the opioid µ1 receptors [30]. Among the latter, the most important are the highs’ greater excitability of the motor cortex [31] and stronger functional equivalence (FE) between actual and imagined perception/action [32]. These differences make hypnotizability a good predictor of the efficacy of motor imagery [12].

Hypnotizability-related brain differences also include reduced grey matter volume in the insula and in the left cerebellum of highs compared to lows [33,34]. Accordingly, highs exhibit lower interoceptive accuracy [35,36] than lows, but »more adaptive« interoceptive sensitivity (IS)—that is, the tendency to trust their bodily signals and behave accordingly—compared to lows and mediums [37]. 

### 1.3. The Aim of This Study

Since both the efficacy of MI [32] and interoceptive sensitivity [37] differ according to hypnotizability, this study aimed to assess the role of hypnotizability and IS in the subjective experience, chronometric index and EEG correlates of visual and kinesthetic motor imagery in healthy highs, mediums and lows. 

We hypothesize that interoceptive sensitivity sustains at least part of the EEG correlates of motor imagery, and that this occurs in highs and lows differentially. The expected results will highlight the role of interoceptive sensitivity in motor imagery for the first time.

## 2. Materials and Methods

### 2.1. Participants

Fifty-three right-handed (according to the Edinburgh Handedness Inventory) healthy volunteers of both sexes (age range: 19–26 years) were recruited among the students at the University of Pisa. After signing the informed consent approved by the University Bioethics Committee (n.29/2022), the absence of medical, neurological, psychiatric disease, sleep and attention disorders and current pharmacological therapies was assessed by anamnestic interviews. Then, the participants were administered the Italian version of the Stanford Hypnotic Susceptibility Scale: Form A (SHSS: A, range: 0–12) [38], classifying them into groups of highs (score > 8 out of 12), mediums (score 5–7) and lows (score < 4). The SHSS scale consists of 12 behavioral items (for instance, postural instability, eyes closure, arms heaviness, rigidity and immobilization, hallucination of a mosquito, post-hypnotic command, amnesia). Each item is marked as passed/not passed depending on whether the hypnotist sees behavioral responses of fixed magnitude to suggestions within 10 s from the end of the suggestion. In the general population, mediums represent 70%, while highs and lows each represent 15% [39].

### 2.2. Experimental Design and Procedure

To prevent expectancy effects, questionnaire administration and the experimental session were performed at least two weeks after hypnotic assessment. The proneness to be deeply involved in cognitive tasks and interoceptive sensitivity were assessed through the Tellegen Absorption Scale (TAS, [40]) and the Multidimensional Assessment of Interoceptive Awareness (MAIA, [20]). The experiments were conducted between 9 and 12 a.m. in a semi-dark and quiet room. Participants sat in a comfortable armchair with their eyes closed. Before the experimental session started, one of the experimenters demonstrated the movement to be performed and later imagined. The movement consisted of ten repetitions of flexion–extension of the left arm and hand, along with finger-to-thumb tapping. Its execution was visually inspected by an experimenter.

The experimental session consisted of a 2 min resting state (baseline), followed by two series of actual and imagined movements, performed through two different modalities (visual and kinesthetic). An actual movement (Mv) preceded each series of three repetitions of the same, visually imagined (MIv) and kinesthetically imagined movement (MIk). Each imagery series was also preceded by listening to a recorded script describing the movement (Appendix A). For visual imagery, the script read, “…*Now please imagine doing the same movement you did a few minutes ago. You can see your left arm flexing up to the shoulder while your fingers touch your thumb one by one from the index to the little finger…*”. For the kinesthetic imagery, the script read, “…*Now, please imagine repeating the same movement you did a few minutes ago. You can feel the tension growing in your left biceps as it flexes up to the shoulder, while the muscles in your forearm start contracting and your fingers touch your thumb one by one from the index to the little finger …*”.

The conditions were interspersed with 30 s rest periods. The duration of each condition was monitored. The imaginative visual and kinesthetic sequences were randomized among the subjects of each group (Figure 1). Each condition was initiated by a vocal command and ended when the experimenter saw that the real movement had ended (Mv, Mk), or when the subject declared that the imagery task had been completed (VI, KI). At the end of each imagery condition, the participants rated the experienced efficacy of imagery (range: 0–10).

### 2.3. Signal Acquisition and Analysis

The electrocardiogram (ECG) and electroencephalogram (EEG) were acquired by the g.tec’ smultipurpose wireless biosignal acquisition tool g. Nautilus (G.tec Medical Engineering, Graz, Austria). An EEG cap with 28 electrodes placed in standardized positions according to the modified 10-10 international system was used (FP1, FP2, AF3, AF4, F7, F3, Fz, F4, F8, C3, FC1, FC2, C4, T7, Cz, T8, CP5, CP1, CP2, CP6, P7, P3, Pz, P4, P8, PO3, PO4, Oz) and the reference was set to Cz. Two additional ECG electrodes were placed underneath the right and left clavicle. The sampling rate was set to 500 Hz and all impedances were kept below 30 kΩ.

EEG pre-processing was performed with the EEGLAB toolbox [41]. Signals were band-pass filtered (band pass: 0.5–45 Hz, two-way least-squares FIR filtering, according to the frequency bands analyzed, 8–33 Hz) and visually inspected to reject physiological and non-physiological artifacts. Individual channels showing quality decline (due to instability or loss of contact with the scalp during recordings) were visually identified and replaced with signals obtained via spline interpolation, a method generally implemented to maintain the spectral characteristics of the signal [42]. To remove residual artifacts, values exceeding the range of −70–70 mV were discarded. The retained EEG signals were downsampled to 256 Hz to reduce the required computing power. The signal was then submitted to Independent Component Analysis (ICA, [43]) to remove ocular, heart and muscular artifacts in each subject. We used the extended ICA algorithm and selected 20 components explaining the highest variance, since the optimal number of components should be between half and three quarters of the electrode number, thus balancing between identifying sufficient variance and avoiding overfitting [41]. Artifact components were identified by visual inspection of their time course, power spectrum and scalp maps. For every condition (B, M, MIv and MIk), the pre-processed EEG signal was subjected to Power Spectral Density (PSD) analysis using Welch’s method. PSD estimation was performed with a Hamming window of 4 s length, as a compromise between frequency and temporal resolution, and 50% overlap, to reduce spectral variability. The choice of using a Hamming window rather than a rectangular window was justified by the necessity to avoid spectral distortions due to border discontinuity. For F3, F4, C3, C4, P3, P4, PO3, PO4 channels, the PSD was integrated over the alpha (8–12 Hz), low-beta (13–21 Hz) and high-beta (22–30 Hz) frequency bands and then log-transformed. PSD was averaged across the three visual and across the three kinesthetic imagery conditions, and across the two actual movement conditions. Signals from frontal and central, as well as parietal and occipital electrodes were averaged to obtain two regions (fronto–central, FC, and parieto–occipital, PO) for every condition. The heart rate was extracted by the open-source MATLAB toolbox EEG-Beats [44], which downsamples the signal to 128 Hz, applies FIR filtering (3 Hz high-pass, 20 Hz low-pass) and then uses a divide-and-conquer strategy to identify ECG peaks.

### 2.4. Variables

#### 2.4.1. Preliminary Evaluation

##### Hypnotic Assessment

According to the SHSS: A, the participants were 16 highs (*M* + *SD*: 9.75 + 1.24; 9 females), 11 mediums (6.00 + 0.77; 7 females), 26 lows (1.00 + 1.52; 13 females), with mediums representing 70% of the general population and highs and lows each representing 15% [39]. SHSS scores were significantly different across the hypnotizability groups (*F*(2, 50) = 224.90, *p* < 0.001), with the highs’ scores higher than those of the mediums (*p* < 0.001) and lows (*p* < 0.001) and the mediums’ scores higher than those of the lows (*p* < 0.001).

##### Questionnaires

-Absorption (Tellegen Absorption Scale (TAS, [40]), measuring the level of proneness to be deeply involved in cognitive tasks (range: 0–34).-Interoceptive sensitivity (Multidimensional Assessment of Interoceptive Awareness (MAIA [20]), consisting of 8 dimensions (noticing, not distracting, not worrying, attention regulation, emotional awareness, self-regulation, body listening, trusting), with each item scored between 0 (min) and 5 (max).

#### 2.4.2. Experimental Session

-For physiological assessment: EEG alpha and beta PSD and heart rate (HR) during the baseline condition (B), actual movement (M), visual and kinesthetic MI (VI, KI).-For psychophysiological assessment, visual (ΔV) and kinesthetic (ΔK) imagery duration normalized to movement duration computed as [(actual movement duration—imagined movement duration)/actual movement duration].-For the subjective experience of MI, self-report of the efficacy of visual (Ve) and kinesthetic (Ke) imagery (range: 0–10).

### 2.5. Statistical Analysis

To assess hypnotizability group differences, a univariate ANOVA was used for TAS scores and a multivariate ANOVA was applied to MAIA dimensions. Separate repeated measures ANOVAs (2 modalities × 3 groups) were applied to ΔHR, chronometry and efficacy variables. EEG log transformed signals were studied by repeated measures ANOVAs according to the following experimental design: 3 groups (highs, mediums, lows) × 4 conditions (B, M, V, K) × 2 hemispheric sides (left, right) × 2 brain regions (fronto–central, parieto–occipital).

The Greenhouse–Geisser correction was used when the sphericity assumption was not met. ANCOVA was applied to test the differences in chronometry, efficacy and EEG using MAIA dimensions as covariates. Post-hoc comparisons were performed by contrast analysis and paired *t*-tests. Bonferroni correction for multiple comparisons was applied when necessary. Spearman correlation was used to study associations between chronometric, subjective measures of efficacy of each imagery modality and EEG alpha-/low-beta/high-beta power spectra. The significance level was set at α = 0.05.

## 3. Results

All the enrolled participants completed the study. Table 1 reports the mean values and standard deviations of all the studied variables. The TAS scores were significantly different between hypnotizability groups (*F*(2, 50) = 6.99, *p* = 0.002), with the highs’ (*p* = 0.006) and mediums’ scores (*p* = 0.019) higher than those of the lows, and no difference between highs and mediums. Among the MAIA dimensions of interoceptive sensitivity, only noticing differed among the hypnotizability groups (*F*(2, 50) = 3.86, *p* = 0.028), with highs’ scores > lows’ (*p* = 0.046).

### 3.1. Motor Imagery, Hypnotizability and Interoception 

#### 3.1.1. Heart Rate

Repeated measures ANOVA did not reveal a significant group effect for ΔHR. The changes in heart rate between visual imagery and actual movement (ΔHRv) were larger than the changes occurring between kinesthetic imagery and actual movement (ΔHRk) (*F*(1, 50) = 28.63, *p* < 0.001). 

#### 3.1.2. Subjective Efficacy (Ve, Ke)

Repeated measures ANOVA (3 groups × 2 MI modalities) revealed significant differences between imagery modalities (K > V, *F*(1, 50) = 16.82, *p* < 0.001, η^2^ = 0.252, α = 0.980) and a significant modality × group interaction (*F*(2, 50) = 3.918, *p* = 0.026, η^2^ = 0.135, α = 0.680). Its decomposition revealed a significant difference between imagery modalities in highs (K > V, *F*(1, 15) = 6.79, *p* = 0.020) and mediums (K > V, *F*(1, 10) = 7.56, *p* = 0.02) and no difference in lows. The kinesthetic efficacy was greater in mediums compared to lows (*p* = 0.046) and did not differ between highs and mediums and between highs and lows. The visual efficacy was similar in all three groups (Figure 2a). Controlling for MAIA dimensions in the ANOVA abolished all differences. 

#### 3.1.3. Chronometry (ΔV, ΔK)

A repeated measures ANOVA (3 groups × 2 MI modalities) did not reveal significant differences between groups and imagery modalities (Figure 2b). No difference was found by controlling for MAIA dimensions in the ANOVA.

#### 3.1.4. Correlational Analysis

Chronometric (ΔK, ΔV) and subjective (Ke, Ve) measures of motor imagery did not correlate with each other for both modalities of imagery. Partial correlation controlling for SHSS and MAIA dimensions did not show any significant difference.

### 3.2. EEG Alpha and Beta PSD 

Based on both the functional equivalence model [11] and on the literature showing lower desynchronization during imagery than during actual movement [45,46], we studied the conditions in which K and/or V power spectra were higher or non-significantly different from M, given the presence of significantly greater desynchronization during movement and imagery compared to baseline conditions.

The EEG power spectra of alpha, low-beta and high-beta showed significant differences between the baseline (B) and movement (M) conditions at all sites. 

Alpha PSD (Figure 3)

**Figure 3 brainsci-14-00832-f003:**
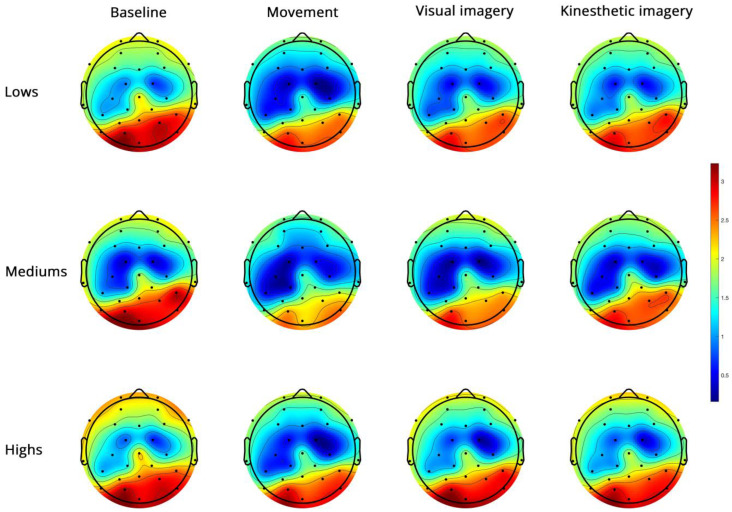
Topoplots of EEG alpha PSD during baseline, real movement, visual and kinesthetic imagery in lows (upper panel), mediums (middle panel) and highs (lower panel).

The side and condition effects were abolished by controlling for the MAIA dimensions. The region effect (*F*(1, 42) = 6.62, *p* = 0.014, η^2^ = 0.136 α = 0.710) and the region × condition × group interaction (*F*(6, 126) = 3.05, *p* = 0.013, η^2^ = 0.126 α = 0.854) survived. In highs, the decomposition of this interaction revealed significant region (FC < PO) and condition effects, which, however, did not involve differences between B, V and K (Appendix A). The significant main effects and interactions are reported in Table 2.

In mediums, Table 2 shows significant side, region and condition effects with M = V and M = K also observed (Figure 4, Appendix A).

In lows, the region × condition interaction revealed FC power < PO power in all conditions, B = K = V in FC, and M < K in PO (Appendix A).

No significant group effect was observed.

Low-beta PSD (Figure 5).

**Figure 5 brainsci-14-00832-f005:**
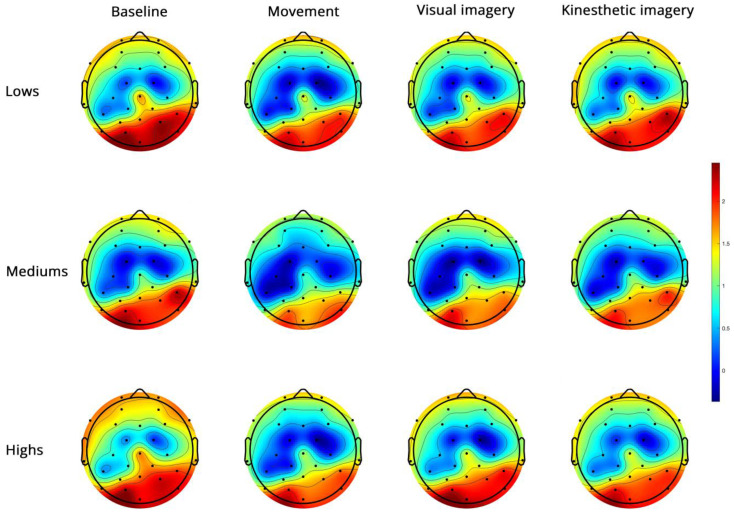
Topoplots of EEG low-beta PSD during baseline, real movement, visual and kinesthetic imagery in lows (upper panel), mediums (middle panel) and highs (lower panel).

Low-beta showed significantly higher power in the right hemisphere, in the PO region and during V and K compared to M (Appendix A).

Table 3 shows the main effects and interactions for low and high-beta.

High-beta PSD (Figure 6).

**Figure 6 brainsci-14-00832-f006:**
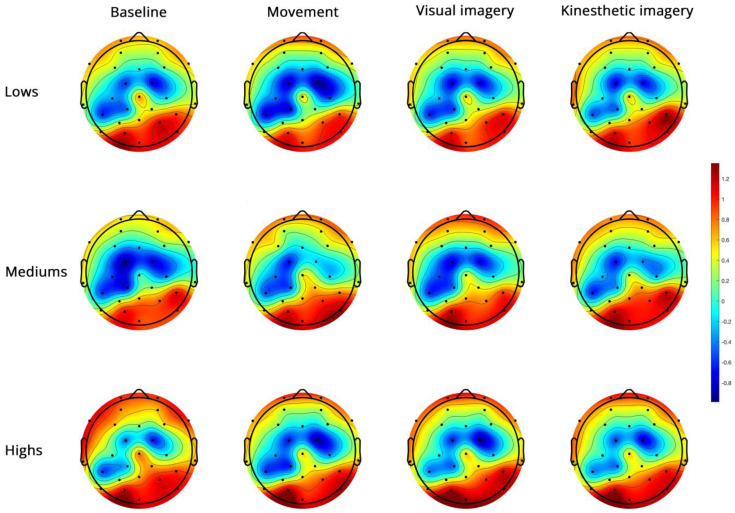
Topoplots of EEG high-beta PSD during baseline, real movement, visual and kinesthetic imagery in lows (upper panel), mediums (middle panel) and highs (lower panel).

For high-beta (Figure 6), the significant side x group interaction (Table 3) revealed a significant side effect only in mediums (Appendix A). The significant region x condition interaction (Figure 7) indicated FC power < PO in all conditions, M < K in FC and no difference between conditions in PO (Appendix A).

All main effects and interactions described for low-beta and high-beta (Figure 7) became non-significant when controlling for MAIA.

No significant correlation was observed between EEG alpha/beta PSD and subjective or chronometric variables.

## 4. Discussion

This study aimed to investigate the role of hypnotizability and interoceptive sensitivity in motor imagery performed through the visual or kinesthetic sensory modality. The findings enable us to report, for the first time, a relevant role of interoceptive sensitivity in the subjective experience of movement imagery and its cortical correlates. In contrast, chronometric indices were not influenced by IS. Only alpha and high-beta PSD displayed a few hypnotizability-related differences.

### 4.1. Subjective and Behavioral Findings

The findings show that both interoceptive sensitivity and hypnotizability contribute to the difference in the subjective experience of visual and kinesthetic MI among the three groups, but not to their chronometric differences. The absence of a correlation between chronometry and subjective experience in both modalities of imagery was not unexpected, owing to hypnotizability-related differences in imagery abilities [32,47] and interoceptive characteristics [37], which might buffer each other differentially across the three groups. Subjective experience and chronometric indices of MI have been described, however, as measures of different components of motor imagery, whose effects may not be necessarily correlated with each other [48]. 

The subjective reports of the efficacy of visual imagery did not differ between groups, possibly due to the easiness of this modality of imagery. In a previous study, visual imagery was reported as easier than somesthetic imagery by lows when in standing position [49], thus possibly buffering the gap with highs, whereas highs and lows reported the same vividness and effort when seated [50].

Kinesthetic imagery was reported as more effective than visual imagery by mediums and highs, but not by lows, as previously observed in standing participants with different hypnotizability [49] and elite athletes not characterized by hypnotizability [48]. The proneness to absorption, higher in mediums and highs than in lows, might account for the difference between lows and mediums/highs. 

As earlier shown in the general population [49], and in line with the suggested multivariate model of motor imagery [16], no group difference was found in chronometry, despite the hypnotizability-related differences in the efficacy of kinesthetic imagery.

The evaluation of heart rate made the results more robust, as the efficacy of MI requires a multidimensional approach, including autonomic activation (heart rate, electrodermal activity), subjective reports, EEG signals and chronometric measures [16].

### 4.2. EEG Findings

The EEG findings indicate that cortical activities—alpha, low-beta and high-beta—were largely sustained by interoceptive sensitivity, as controlling for the MAIA dimensions in the repeated measures ANOVAs abolished almost all the significant effects. This provides additional insight into the relationship between cognitive functions, such as imagery, and the perception and interpretation of bodily sensations, supporting the concept of embodied cognition [51].

EEG alpha spectral frequency, whose changes were sustained by interoceptive sensitivity, highlighted three different, hypnotizability-related modes of cortical elaboration of the imagery tasks. In highs, the cortical activities associated with both imagery tasks did not differ from the baseline, despite their subjective and behavioral correlates. This could be a side effect of the highs’ distributed rather than nested cortical activity, as indicated by earlier studies [32,50,52], which reported relatively few spectral changes in brain activities during sensory–cognitive imagery tasks in this group. Mediums, who represent the general population [39], exhibited the expected desynchronization [7,8], and lows desynchronized only in the parietal–occipital region, suggesting a visual rather than kinesthetic mechanism of motor imagery, independently of the requested modality. This group, similar to the highs, had the same frontocentral alpha power in baseline and imagery conditions. The interpretation of this finding in lows, however, is different from highs, as they could have paid less attention to the task, in line with their lower scores of absorption (TAS).

Low-beta and high-beta changes were sustained by interoceptive sensitivity, too. Low-beta changes (showing lower desynchronization during visual and kinesthetic imagery compared to the actual movement) were observed across the entire sample, as reported for the general population [53], and were sustained by IS. High-beta was modulated during both the kinesthetic and visual imagery in frontocentral regions and, counterintuitively, only during kinesthetic imagery at parieto–occipital sites. Nonetheless, this could be accounted for by the role suggested for beta as a large-scale communication mechanism between sensorimotor areas and other brain regions [54], as well as by the reported similar brain activations during visual and kinesthetic imagery in the general population [55]. The absence of cortical response to visual imagery suggests that, in line with the subjective experience reported by highs and mediums, kinesthetic motor imagery is more effective compared to visual motor imagery [56]. Indeed, the correlates of kinesthetic imagery are more similar to movement, whereas visual correlates better resemble action observation [9]. 

For high-beta, mediums exhibited hemispheric differences consisting of larger left than right desynchronization. In baseline conditions, highs have been reported to have higher cortical activity compared to lows in the frontal left region [57,58], and it is possible that in the present study, the bilateral activations reported for mental imagery [59] buffered the hemispheric difference related to cardiac asymmetric information in highs, and no differences were disclosed in lows. Nonetheless, the mediums’ predominant left activation could suggest a mechanism similar to that of the highs. As a matter of fact, few studies have enrolled mediums; thus, information about their difference from highs and lows is scarce [60]. 

As previously reported [16], and in line with the several discrepancies observed in behavioral and neural correlates of imagined and executed actions [45], no significant correlations were observed between subjective, chronometric and EEG correlates of the studied tasks. Regarding EEG, the debate should also consider the substantial absence of reliable indicators of covert cortical activities [61] and the different styles of information processing observed in participants with different hypnotizability [32,50]. Different methods of EEG signal analysis may likely provide further information. Furthermore, we might argue that the temporal equivalence between actual and imagined movements, which is expressed by the chronometric index, is not necessarily an index of performance accuracy. Indeed, a single reliable index has not been found and we agree with Collet and colleagues [16] on the necessity of a multidimensional evaluation of motor imagery: autonomic responses as indices of task-related arousal, EEG activities as indices of functional equivalence, reported efficacy as subjective experience and chronometry as an index of behavioral correspondence of imagined movement with the actual one. 

#### Limitations

The limitations of this study are the absence of reports about how much participants imagined through the requested sensory modality, and the distribution of the participants’ hypnotizability, which was not in line with the hypnotizability distribution observed in the general population, which includes 15% of highs and lows and 70% of mediums [39]. 

Moreover, the experimental design included series of ten consecutive actual/imagined movements with no marker of each flexion/extension (impossible during imagery). Thus, we were unable to show EEG (de)synchronization for each movement within the series. The desynchronization observed during the entire movement/imagery tasks with respect to the baseline is a consequence of the predominant desynchronization associated with preparation and execution with respect to the synchronization associated with movement termination [5,6].

A strength of this study lies in its protocol, which involved a complex flexion–extension movement of the entire arm, rather than partially automatic, sequential movements limited to hands or fingers [62]. Moreover, the participants were seated rather than lying down [63,64], which was more suitable for performing the studied movement. Finally, the similar EEG correlates of visual and kinesthetic imagery could be accounted for by the fact that the two modalities of imagery were not exclusively used when participants were invited to use one of them, in line with earlier findings of imaging studies [56,64].

## 5. Conclusions

This is the first report of a major role of interoceptive sensitivity in the cortical activities associated with motor imagery. It shows hypnotizability-related differences in the influence of interoceptive sensitivity on motor imagery. From a general perspective, it highlights the close relationship between the individual experience of interoceptive information and cognitive activities, extending to motor imagery the role of the insula, in the light of the concept of embodied cognition. Our findings can be relevant to personalized neurorehabilitation training and to the improvement of sports performance by imagery training. The variable effects of BCI interventions [2,65,66] in post-stroke rehabilitation, in fact, could be attributed either to ineffective motor imagery or to altered interoceptive sensitivity. Moreover, since motor imagery training has also been found effective in improving the patients’ emotional experience and motivation to achieve a better quality of life [2], a holistic evaluation of candidates to BCI interventions, including both interoceptive sensitivity and mental imagery abilities, could better predict the intervention outcome.

## Figures and Tables

**Figure 1 brainsci-14-00832-f001:**
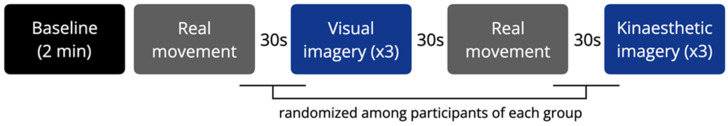
Experimental design of the study. Blocks represent the experimental conditions, which were separated by a 30 s rest. Each imagery condition was repeated 3 times.

**Figure 2 brainsci-14-00832-f002:**
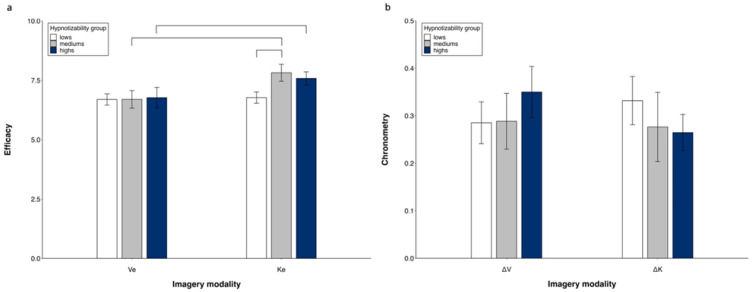
Visual and kinesthetic imagery. Subjectively reported efficacy (**a**) and chronometric values (**b**). Mean values and standard errors. Lines indicate significant differences.

**Figure 4 brainsci-14-00832-f004:**
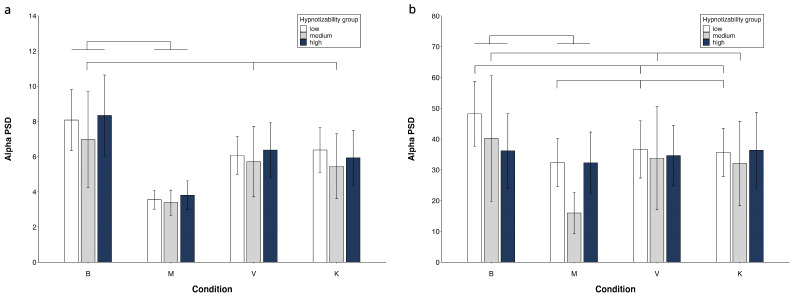
Alpha PSD. Region × condition × group interaction. FC region (**a**) and PO region (**b**). Mean values and standard errors. Lines indicate significant differences (differences between M and V/K are presented only if V/K are different from B).

**Figure 7 brainsci-14-00832-f007:**
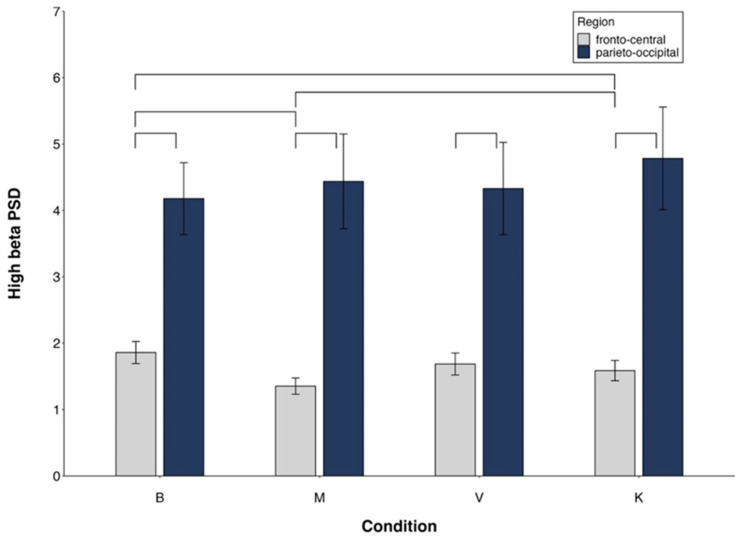
High-beta PSD. Region × condition interaction. Mean values and standard errors. Lines indicate significant differences (differences between M and V/K are presented only if V/K are different from B).

**Table 1 brainsci-14-00832-t001:** Descriptive statistics of the studied variables.

		Lows		Mediums		Highs	
		M	SD	M	SD	M	SD
TAS *		19.55	5.26	24.27	4.05	24.31	3.75
MAIA	Noticing *	3.18	0.66	3.67	0.78	3.72	0.63
	not distracting	1.94	0.60	1.76	0.53	2.02	1.01
	not worrying	2.76	0.85	2.44	1.33	2.46	0.92
	attention regulation	3.08	0.83	3.20	0.67	3.35	0.54
	emotional awareness	3.44	0.80	3.66	0.94	4.01	0.63
	self-regulation	2.86	0.86	3.22	0.79	2.81	0.87
	body listening	2.56	0.78	2.99	0.93	2.96	0.80
	trusting	3.48	1.11	3.79	1.02	3.37	1.17
ΔHRv		6.41	3.51	8.53	4.94	7.05	6.11
ΔHRk		3.37	2.61	3.44	2.06	4.06	3.68
ΔV		0.28	0.22	0.29	0.19	0.35	0.22
ΔK		0.33	0.25	0.28	0.24	0.26	0.15
Ve		6.68	1.16	6.70	1.22	6.77	1.72
Ke *		6.78	1.17	7.82	1.19	7.58	1.09

Note: * significant differences among hypnotizability groups. For details, see the text.

**Table 2 brainsci-14-00832-t002:** ANOVA results for alpha PSD.

Effect		*F*	df	*p*	η^2^	α	MAIA
side		4.09	1, 50	0.048	0.754	0.510	ns
region		273.89	1, 50	0.0001	0.846	0.999	
condition		18.74	3, 150	0.0001	0.273	0.999	ns
region x condition x group		2.35	6, 150	0.048	0.086	0.618	
with:	**highs**						
region		103.68	1, 15	0.0001	FC < PO		
condition		4.04	3, 45	0.038			
	**mediums**						
region		48.22	1, 10	0.0001	FC < PO		
condition		6.37	3, 30	0.005			
	**lows**						
region		177.96	1, 25	0.0001	FC < PO		
condition		12.90	3, 75	0.0001			
region x condition		3.16	3, 75	0.042			

**Table 3 brainsci-14-00832-t003:** ANOVA results for low and high-beta PSD.

Effect	Low-Beta	*F*	df	*p*	η^2^	α		MAIA
side		7.28	1, 50	0.009	0.127	0.754	left < right	ns
region		360.73	1, 50	0.0001	0.877	0.999	FC < PO	ns
condition		37.81	3, 150	0.0001	0.431	0.999		ns
	**high-beta**							
side		12.79	1, 50	0.0001	0.204	0.939	left < right	ns
region		196.40	1, 50	0.0001	0.797	0.999	FC < PO	ns
side x group		3.79	2, 50	0.029	0.131	0.664		ns
condition		6.46	3, 150	0.001	0.114	0.951		ns
region x condition		3.72	3, 150	0.022	0.069	0.715		ns

## Data Availability

The original data presented in this study are available upon request.

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
