# Peer review of "The Role of Interoceptive Sensitivity and Hypnotizability in Motor Imagery"

_brainsci, 2024, doi:10.3390/brainsci14080832_

Round 1

Reviewer 1 Report

Comments and Suggestions for Authors

Thank you for the opportunity to review the article with the title - The role of interoceptive sensitivity and hypnotizability in motor imagery. The article is interesting and addresses a topical issue.

Recommendations:

Introduction

- to detail the new aspects of the study in relation to previous studies on the same topic.

- at the end of the section to add the hypothesis of the study.

Study design - to add the periods of the study and phasing them in time.

The results are well structured and clearly interpreted.

The limits of the study and to be included in the Discussions are separated.

In Discussions - to detail the practical implications.

Conclusions to be included in a separate section.

Author Response

- to detail the new aspects of the study in relation to previous studies on the same topic.:

Line 70 Nonetheless, no study has been performed on the role of interoceptive sensitivity in motor imagery, despite the well-recognized role of the insula, whose activity is associated with both interoceptive accuracy and sensitivity( )  in integrating interoceptive and exteroceptive signals.

- at the end of the section to add the hypothesis of the study.:

Line 85-86 We hypothesize that interoceptive sensitivity sustains at least part of the EEG correlates of motor imagery. 

Study design - to add the periods of the study and phasing them in time.:

 Added: between 9 an 12a.m.

The results are well structured and clearly interpreted.

The limits of the study and to be included in the Discussions are separated.

Limitations and conclusions have been separated

In Discussions - to detail the practical implications.

DONE

Conclusions to be included in a separate section

DONE.

Reviewer 2 Report

Comments and Suggestions for Authors

This study used electroencephalography to investigate the relationship between proprioceptive sensation and hypnotic susceptibility in motor imagery. The idea behind this study is interesting. However, I have some concerns, so I have commented on it.  

1. p1, line 32. The idea for this research is interesting. The author should probably explain hypnotizability in more detail.

2. p3, line 96 This is results. The author should rewrite this text as a results section. In addition, the classification using SHSS needs to be explained in more detail.

3. p3, line 104 What is movement task?

4. p3, Participants performed motor imagery after the actual movement. Investigating the vividness of motor imagery is an important indicator of whether or not motor imagery was successful. Why didn't the author ask about the vividness of motor imagery?  

5. p4, line 153, band-pass filter applied 0.5-45Hz. Please clarify why you selected this filter.  

6. p4, The motor imagery was performed for 30 seconds. Epoching could be taken into account, but why didn't the author apply epoching? Was it to have an influence on ICA?

7. There are many blank spaces in the reference section. Please fill them in.

Comments on the Quality of English Language

No comment.

Author Response

This study used electroencephalography to investigate the relationship between proprioceptive sensation and hypnotic susceptibility in motor imagery. The idea behind this study is interesting. However, I have some concerns, so I have commented on it.  

p1, line 32. The idea for this research is interesting. The author should probably explain hypnotizability in more detail.

DONE: Hypnotizability is a psychophysiological trait quite stable through life (Piccione etc, 1989), and is mostly known for the ability of high hypnotizable individuals to control pain through cognitive strategies (   ). It is measured by standardized scales and is associated with several behavioral and brain morpho-functional characteristics [4]. Among them, different habituation of spinal reflexes, greater excitability of the motor cortex, better vascular response to mental stress and nociceptive stimulation, lower difference between the EEG correlates of actual and imagined movement (Functional equivalence, FE) ,  different sensitivity of opiates mu1 receptors (Santarcangelo, 2024 LIBRO). Based on behavioral and EEG studies, some of these differences make the hypnotizability score a good predictor of the efficacy of motor imagery (9 ) , i.e. highly hypnotizables’ stronger functional equivalence etween actual and imagined sensorimotor conditions [7,8], which is defined as stronger functional equivalence (FE) [9], and the higher excitability of the motor cortex ( ) 

  1. p3, line 96 This is results. The author should rewrite this text as a results section. In addition, the classification using SHSS needs to be explained in more detail.

Done: The SHSS scale consists of 12 behavioral items  (for instance, postural instability, eyes closure, arms heaviness, rigidity and immobilization, hallucination of a mosquito, post-hypnotic command,amnesia). Each item is passed/not passed whether the hypnotist sees behavioral responses of fixed largeness to suggestions within 10 seconds from the end of the suggestion

  1. p3, line 104 What is movement task?

This was described in the Experimental procedure: The movement consisted of ten repetitions of flexion-extension of the left arm, hand, and fingers to touch the thumb.

  1. p3, Participants performed motor imagery after the actual movement. Investigating the vividness of motor imagery is an important indicator of whether or not motor imagery was successful. Why didn't the author ask about the vividness of motor imagery?  

The vividness of motor imagery has been called “subjective efficacy”

  1. p4, line 153, band-pass filter applied 0.5-45Hz. Please clarify why you selected this filter. 

 . We filtered just to avoid high frequency muscle artifacts and low frequency thermoregulatory , respiratory and other electrical artefacts and to maintain lower and higher frequncies for further analyses

  1. p4, The motor imagery was performed for 30 seconds.

Duration of both actual and imagined movements were different among subjects.

Epoching could be taken into account, but why didn't the author apply epoching? Was it to influence ICA?  

We divided each condition in 4 seconds epochs (50% ovelap)

  1. There are many blank spaces in the reference section. Please fill them in.

DONE

Reviewer 3 Report

Comments and Suggestions for Authors

Title: “The role of interoceptive sensitivity and hypnotizability in motor imagery”

Manuscript ID: brainsci-3107859-peer-review-v1

This work proposes a first analysis of the role of interoceptive sensitivity in the cortical activities associated with motor imagery. Concretely, the close relationship between interoceptive sensitivity and hypnotizability in motor imagery is explored, aiming to shed light on the embodied cognition concept.

The topic is interesting in the fields of neurorehabilitation, brain-computer interface-based applications, and imagery training. However, the developed method could have offered a deep analysis of new insights into the interoceptive sensitivity associated with motor imagery performed through visual and kinesthetic modalities. But this is not the case.

I have some suggestions to improve the quality of the manuscript.

A. Abstract

1. The manuscript's contribution ought to be presented in the present tense rather than the past. I am aware that the experiment has already been done. See lines 11–14 and 15–19 on page 1.

2. As the visual and kinesthetic modalities of the participants were classified according to the hypnotizability scores, it is expected to find in this section not only the literary but also the numerical results.

3. Consider improving for this section the English writing style and spelling.

B. Keywords

1. I suggest that the authors include MAIA among the keywords.

2. It is more suitable to use "interoceptive sensitivity" instead of "interception."

C. Introduction

1. Consider improving this sentence: "It is used in neuro-rehabilitation therapies, including brain-computer interface interventions; however, its efficacy is difficult to predict due to the large variability of motor imagery abilities in the general population." For example, “...interventions. However, its…”, or “...due to the large variability of motor imagery abilities across subjects.”

2. Consider checking the beginning of this sentence: "Actual and effective imagined actions also share time duration..." Something is wrong. See on Page 1, line 39.

3. The introductory section should be rewritten, highlighting the contributions of this work, improving the English writing style and spelling, and logically organizing the sequence of paragraph contents. Reading through the introduction, I do not feel the cohesion between the ideas developed in each paragraph.

4. It would be interesting to add the related work section after the introductory one. This would allow the contributions of this article to be better highlighted.

5. The keywords "Multisensory Assessment of Interoceptive Awareness (MAIA)" and EEG presented in the abstract were left without development or bibliography in the introduction.

D. Materials and Methods

1. According to what is stated on Page 2, lines 90–93, namely, "... with mediums representing 70% of the general population and highs and lows each representing 15% [32], how did the authors find the general population representation of 70%?

2. The results presented from line 93, Page 2, to line 96, Page 3, are not clear. In addition to the expected explanations, consider improving the English writing style and grammar.

3. What is the relevance of this sentence beginning, "The experiments were conducted on the same day..." compared to the previous paragraph and the text that follows?

4. According to what is said on Page 3, from lines 105 to 120, I can suspect that the conductor of the experiment had no software support to confirm that the imagined movements were actually executed. Was only the declaration of the subject, as said on lines 117–118, sufficient to approve the imagined task? In similar experiments in the literature, these crucial moments of capture are monitored to observe clear differences between the cue signals and those of the baseline.

5. The caption of Figure 1 is very silent. Please, add more explanations about each diagram block.

6. This section needs a thorough revision of English grammar and spelling.

7. The text from Page 3, lines 125–132, and then lines 134–143 of Page 4 should be summarized for this section, and its entirety should be moved to the additional material. These instructions for the experiment subjects should be short but concise. Clear and brief.

8. I am surprised to discover electrocardiogram (ECG) as a technique developed by the authors in the manuscript. Neither the abstract nor the introduction mention this technique.

9. Some justifications for setting the sampling frequency at 500 Hz and all impedances below 30 kΩ?

10. Section 2.4, related to signal acquisition and analysis, should be rewritten by presenting step-by-step each technique utilized in the chain processing. See on Page 4. There is already evidence to question the authors' results because of this lack of clarity. In particular, why was the two-way least-squares FIR filtering with a bandwidth of 0.5–45 Hz selected? Although alpha and beta (7–30 Hz) are the bands of interest that are specifically mentioned in the summary section. Why did the authors use the spline interpolation technique to substitute noisy signals from specific channels? The findings to be acquired are tainted by the gathering of additional data since the spline technique—even cubic—has limitations when it comes to extrapolation. The signals were downsampled to 256 Hz; why? With what settings was the ICA algorithm applied? How many independent components were employed for the analysis for each channel

and each subject, and why was that decision made? Why is there another downsampling to 128 Hz?

11. Section 2.6 is expected to present statistical analysis, but it is empty. Please consider including statistical variables, as claimed by the section title.

E. Results

In addition to the doubts, I expressed regarding the findings of the research in observation 10 of the Materials and Methods section, I completely disagree with any findings made in this section for the reasons listed below:

1. The procedure for distinguishing between alpha and beta waves is not rendered clear in the manuscript (refer to Tables 2 and 3, pages 7 and 8).

2. Data used was a combination of data generated by the spline interpolation and those really recorded.

3. The PSD algorithm's spectral analysis lacks enough justification and explanation.

4. The number of subjects is not disclosed in the results that are presented. It's unclear if each experiment subject completed the task accurately.

5. Do the presented averages relate to the total number of sessions, subjects, and number of channels?

6. of the 58 references provided in the manuscript, 5 are incomplete (missing number of pages, confusion of the year of publication, etc.), 3 cannot be found on the publications’ hub, and 16 are old more than 12 years.

7. The quality of the writing in English is below average. The authors should revise the entire manuscript.

8. 13 Self-citations by authors of works.

F. Discussion

1. It is missing a discussion about the ECG-developed technique.

2. Unless I am mistaken, I did not find in the manuscript a comparison of the results presented by the authors with those published in the related literature, nor a rational discussion resulting from them.

G. Limitations and conclusions

1. Avoid putting references in the conclusion because this section summarizes your work by essentially recalling the goals of your work, the method developed, the results obtained, the limitations, and future projections of your work.

2. The limitations presented on page 11, lines 360–363, sufficiently show how the results presented are not robust and consistent. My advice would be to encourage authors to return to deep and methodical testing.

Recommendation: Reject (updates required before a further submission)

Comments on the Quality of English Language

The quality of the writing in English is below average. The authors should revise the entire manuscript.

Author Response

This work proposes a first analysis of the role of interoceptive sensitivity in the cortical activities associated with motor imagery. Concretely, the close relationship between interoceptive sensitivity and hypnotizability in motor imagery is explored, aiming to shed light on the embodied cognition concept.The topic is interesting in the fields of neurorehabilitation, brain-computer interface-based applications, and imagery training. However, the developed method could have offered a deep analysis of new insights into the interoceptive sensitivity associated with motor imagery performed through visual and kinesthetic modalities. But this is not the case.I have some suggestions to improve the quality of the manuscript.

  1. Abstract
  2. The manuscript's contribution ought to be presented in the present tense rather than the past. I am aware that the experiment has already been done. See lines 11–14 and 15–19 on page 1.

Done in the results part of the abs

  1. As the visual and kinesthetic modalities of the participants were classified according to the hypnotizability scores, it is expected to find in this section not only the literary but also the numerical results.

Groups hypnotizability scores have been included

3.Consider improving for this section the English writing style and spelling.

We checked the text

KeywordsI suggest that the authors include MAIA among the keywords. DONE

It is more suitable to use "interoceptive sensitivity" instead of "interception."DONE

  1. Introduction
  2. Consider improving this sentence: "It is used in neuro-rehabilitation therapies, including brain-computer interface interventions; however, its efficacy is difficult to predict due to the large variability of motor imagery abilities in the general population." For example, “...interventions. However, its…”, or “...due to the large variability of motor imagery abilities across subjects.”

DONE

  1. Consider checking the beginning of this sentence: "Actual and effective imagined actions also share time duration..." Something is wrong. See on Page 1, line 39.

DONE

  1. The introductory section should be rewritten, highlighting the contributions of this work, improving the English writing style and spelling, and logically organizing the sequence of paragraph contents. Reading through the introduction, I do not feel the cohesion between the ideas developed in each paragraph.

We have reorganized this section. In the present version of the manuscript, the logic map of the text is: 1)Motor imagery, 2) motor imagery and interoception, 3)hypnotizability and motor image, 4) hypnotizability and interoception

  1. It would be interesting to add the related work section after the introductory one. This would allow the contributions of this article to be better highlighted.

Line 65-67: Nonetheless, no study has been performed on the role of interoceptive sensitivity in motor imagery, despite the well-recognized role of the insula, whose activity is associated with interoceptive accuracy and sensitivity 

The background of the study is presented in the Introduction: We have added, atits end,:  The expected results will highlight for the first time the role of  interoceptive sensitiv ity and hypnotizability  in motor imagery, which is a novel topic

The keywords "Multisensory Assessment of Interoceptive Awareness (MAIA)" and EEG presented in the abstract were left without development or bibliography in the introduction.

Corrected. IS is mainly measured by the Multidimensional Assessment of Interoceptive Awareness (MAIA,  which consists of 8 scales (noticing, not distracting, not worrying, attention regulation, emotional awareness, self-regulation, body listening, trusting ) indicating the mode of the individual interpretation of interoceptive signals

  1. Materials and Methods
  2. According to what is stated on Page 2, lines 90–93, namely, "... with mediums representing 70% of the general population and highs and lows each representing 15% [32], how did the authors find the general population representation of 70%?

The reference De Pascalis et al, 2000 indicates that this information has been obtained in a representative sample including 15% highs, 15 %lows, 70% mediums.

  1. The results presented from line 93, Page 2, to line 96, Page 3, are not clear. In addition to the expected explanations, consider improving the English writing style and grammar.

The text has been updated and moved to the results section according to rev 2 comments

  1. What is the relevance of this sentence beginning, "The experiments were conducted on the same day..." compared to the previous paragraph and the text that follows?

Authors are usually requested to indicate in which part of the day the experiments are performed owing to the several biological circadian cycles. At least 2 weeks after hypnotic assessment (to prevent expectancy effects), questionnaires administration and the experimental session were performed. The proneness to be deeply involved in cognitive tasks and interoceptive sensitivity were assessed through the Tellegen Absorption Scale (TAS, [33]) and the Multidimensional Assessment of Interoceptive Awareness (MAIA, [34]). the experiments were conducted between 9 an 12a.m. in a semi-dark and quiet room.

  1. According to what is said on Page 3, from lines 105 to 120, I can suspect that the conductor of the experiment had no software support to confirm that the imagined movements were actually executed. Was only the declaration of the subject, as said on lines 117–118, sufficient to approve the imagined task? In similar experiments in the literature, these crucial moments of capture are monitored to observe clear differences between the cue signals and those of the baseline.

In the results section of the original version of the paper, we report the criteria which enabled us to consider the imagery tasks performed according to the literature. “...Based on both the functional equivalence model and on the literature showing lower desynchronization during imagery than during actual movement [38,39], we studied the conditions in which K and/or V power spectra were higher or non-significantly different from M, provided the presence of significant differences between B and V/K,

  1. The caption of Figure 1 is very silent. Please, add more explanations about each diagram block.

Figure 1. Experimental design of the study. Blocks represent the experimental conditions, which were separated by a 30 sec rest.Each imagery condition was repeated 3 times.

  1. This section needs a thorough revision of English grammar and spelling. DONE
  2. The text from Page 3, lines 125–132, and then lines 134–143 of Page 4 should be summarized for this section, and its entirety should be moved to the additional material. These instructions for the experiment subjects should be short but concise. Clear and brief.

Each imagery series was preceded by listening to the script describing the movement (Appendix A, Suppl El Mat). For Visual imagery the suggestions read:“…Now please imagine doing the same movement you did a few minutes ago. You can clearly see your left arm flexing up to the shoulder while your fingers touch your thumb one by one from the index to the little finger…”. For the kinesthetic imagery the suggestions read : “…Now, please imagine repeating the same movement you did a few minutes ago. You can feel the tension growing in your left biceps as it flexes up to the shoulder, while the muscles in your forearm start contracting and your fingers touch your thumb one by one from the index to the little finger …”.

  1. I am surprised to discover electrocardiogram (ECG) as a technique developed by the authors in the manuscript. Neither the abstract nor the introduction mention this technique.

Adding the evaluation of heart rate during cognitive tasks like motor imagery is suggested by Collet et al., 2011(cited in the paper introduction) in the light of the multidimensional assessment of motor imagery. Indeed, The efficacy of MI includes autonomic activation (heart rate, electrodermal activity), subjective reports, EEG signals and chronometric measures.

  1. Some justifications for setting the sampling frequency at 500 Hz and all impedances below 30 kΩ? This is the best setting offered by our recording system. High sampling frequency is often used in literature to improve the temporal resolution of high frequency bands. In addition, the signals will be used for further analysis with different aims (Cohen M, MIT Press, Cambridge, Massachussetts, 2014)
  2. Section 2.4, related to signal acquisition and analysis, should be rewritten by presenting step-by-step each technique utilized in the chain processing. See on Page 4.

This is indeed presented in the text.

There is already evidence to question the authors' results because of this lack of clarity. In particular, why was the two-way least-squares FIR filtering with a bandwidth of 0.5–45 Hz selected? Although alpha and beta (7–30 Hz) are the bands of interest that are specifically mentioned in the summary section. We filtered just to avoid high frequency muscle artifacts and low frequency thermoregulatory , respiratory and other electrical artefacts and to maintain lower and higher frequncies for further analyses

Why did the authors use the spline interpolation technique to substitute noisy signals from specific channels?

 According to   Perrin et al., 1989, this method is well fit to mantain the spectral characteristic of signals.

The findings to be acquired are tainted by the gathering of additional data since the spline technique—even cubic—has limitations when it comes to extrapolation. The signals were downsampled to 256 Hz; why?

In the present study this downsampling satisfied our aims and reduced required computing power.

With what settings was the ICA algorithm applied?

We used Extended ICA algorhythm and selected 20 components explaining the highest amount of variance.

 How many independent components were employed for the analysis for each channel and each subject, and why was that decision made? According to Delorme and Makeig (cited in the text) the optimal number of independent components is between half  and three quarters of the electrodes number in order to identify  artifacts without reducing EEG signal information

Why is there another downsampling to 128 Hz?

This part of the analysis was automatically performed using an open-source MATLAB toolbox EEG-Beats [37], which was applied only to the ECG signal

  1. Section 2.6 is expected to present statistical analysis, but it is empty. Please consider including statistical variables, as claimed by the section title.

The statistics paragraph present in the original version of the paper, following the section 2.5.2. Paragraph 2.5 contais the studied variables before the experimental session SHSS, MAIA, TAS) and during the experimental session (EEG alpha, low beta, high beta, subjective reports, chronometric indices, heart rate.

  1. Results

In addition to the doubts, I expressed regarding the findings of the research in observation 10 of the Materials and Methods section, I completely disagree with any findings made in this section for the reasons listed below:

We have clarified all the reviewer’s doubts regarding the following points and already listed in the methods section

The procedure for distinguishing between alpha and beta waves is not rendered clear in the manuscript (refer to Tables 2 and 3, pages 7 and 8).

Two separate tables are presented for alpha and low/high beta ANOVA results. They are indicated in the text. We have updated the tables to improve the clarity

  1. Data used was a combination of data generated by the spline interpolation and those really recorded.

We have clarified all the reviewer’s doubts regarding this point and already listed in the methods section. Moreover, we have checked that none of the interpolated channels was used in the present study (we only used the electrodes F3, F4, C3, C4, P3, P4, PO3, PO4)

  1. The PSD algorithm's spectral analysis lacks enough justification and explanation.This is written in the Methods section. However, we wish to clarify that we used Hamming windowing rather than a rectangular window to avoid spectral distortions due to borders discontinuity, with 4 sec epochs as a compromise between temporal and frequency resolution to obtain an optimal resolution for alpha and beta bands , 50% overlap owing to the necessity to reduce the spectral variability.
  2. The number of subjects is not disclosed in the results that are presented. It's unclear if each experiment subject completed the task accurately.

All participants completed the study. Reported in the text

  1. Do the presented averages relate to the total number of sessions, subjects, and number of channels?

Yes, they do ( 1 session was performed. PSD was studied at F3, F4, C3, C4, P34, P4, PO3, PO4 for each condition.)

“PSD was averaged across the three visual and three kinesthetic condition and across the two actual movement condition. Signals from frontal and central, as well as parietal and occipital electrodes were averaged for every to obtain two regions (fronto-central, FC, and parieto-occipital,PO ) for every condition”

  1. of the 58 references provided in the manuscript, 5 are incomplete (missing number of pages, confusion of the year of publication, etc.), 3 cannot be found on the publications’ hub, and 16 are old more than 12 years, 13 Self-citations by authors of works.All these referencences are required. We corrected the mistakes

Pisa lab, involved in studies of the physiological correlates of hypnotizability, is the only lab in the world addressing this topic. Thus, it is obvious that we built our research based on our earlier research.

  1. The quality of the writing in English is below average. The authors should revise the entire manuscript. DONE
  2. Discussion
  3. It is missing a discussion about the ECG-developed technique.
  4. We cannot discuss this point because we just applied already developed algorithms to simply identify heart rate.
  5. Unless I am mistaken, I did not find in the manuscript a comparison of the results presented by the authors with those published in the related literature, nor a rational discussion resulting from them.

No earlier findings are present in the specific literature. As we mentioned in the text, this study is a novel approach to the relationship between hypnotizability, motor imagery, and interoceptive sensitivity. A more detailed discussion of EEG findings would have confused the aim of the study and was useless to our aim

  1. Limitations and conclusions
  2. Avoid putting references in the conclusion because this section summarizes your work by essentially recalling the goals of your work, the method developed, the results obtained, the limitations, and future projections of your work. DONE
  3. The limitations presented on page 11, lines 360–363, sufficiently show how the results presented are not robust and consistent. My advice would be to encourage authors to return to deep and methodical testing.

We think to have replied to and clarified each point addressed by this reviewer, and to have matched the rev 1 and rev 2 suggestions. 

The quality of the writing in English is below average. The authors should revise the entire manuscript.

Done

Reviewer 4 Report

Comments and Suggestions for Authors

This study provides intriguing insights into the mechanisms of motor imagery. The topic is relevant to the journal audience and the special section.

The paper would benefit from a more comprehensive explanation of the experimental methods. For instance, including at least one figure to illustrate the obtained EEG patterns (ERD/S) for each group/task with confidence intervals should be beneficial.

The findings of this study are particularly interesting in the context of developing new rehabilitation systems based on brain-computer interface, utilizing motor imagery in a closed loop with visual or motor feedback.

Mane, R., Chouhan, T. and Guan, C., 2020. BCI for stroke rehabilitation: motor and beyond. Journal of neural engineering17(4), p.041001.

Recent research indicates that the severity of impairment influences the outcomes of such therapies. It is possible that in severely disabled patients, the ability to perform motor imagery is compromised, which may limit the effectiveness of these devices or necessitate more intensive use, and could methods described here be utilized to predict such outcomes or select effective imagery modality:

Brunner, I., Lundquist, C.B., Pedersen, A.R., Spaich, E.G., Dosen, S. and Savic, A., 2024. Brain computer interface training with motor imagery and functional electrical stimulation for patients with severe upper limb paresis after stroke: a randomized controlled pilot trial. Journal of NeuroEngineering and Rehabilitation, 21(1), p.10.

Further disscussuon of potential implications of this work in such scenarios should be further discussed.

Author Response

This study provides intriguing insights into the mechanisms of motor imagery. The topic is relevant to the journal audience and the special section.

The paper would benefit from a more comprehensive explanation of the experimental methods. For instance, including at least one figure to illustrate the obtained EEG patterns (ERD/S) for each group/task with confidence intervals should be beneficial.

We are unable to include the event-related (de)synchronization plots in the manuscript as in our research design we did not consider each performed / imagined movement as a separate event within the ten movements series. Due to the large variability in the movement / imagery duration (which was used to calculate the chronometric index) we also could not treat the entire sequence of movements as a single event and average the evoked responses to obtain group-level results.

We have included this consideration in the limitation section

( … the experimental design included series of ten consecutive actual/imagined movements with no marker of each flexion/extension (impossible during imagery). Thus, we were unable to show EEG (de) synchronization for each movement within the series. The desynchronization observed during the entire movement/imagery tasks with respect to baseline is a consequence of the pre-eminent desynchronization associated with preparation and execution with respect to the synchronization associated with the movements termination [5,6]...)

The findings of this study are particularly interesting in the context of developing new rehabilitation systems based on brain-computer interface, utilizing motor imagery in a closed loop with visual or motor feedback.

Mane, R., Chouhan, T. and Guan, C., 2020. BCI for stroke rehabilitation: motor and beyond. Journal of neural engineering, 17(4), p.041001.

Recent research indicates that the severity of impairment influences the outcomes of such therapies. It is possible that in severely disabled patients, the ability to perform motor imagery is compromised, which may limit the effectiveness of these devices or necessitate more intensive use, and could methods described here be utilized to predict such outcomes or select effective imagery modality:

Brunner, I., Lundquist, C.B., Pedersen, A.R., Spaich, E.G., Dosen, S. and Savic, A., 2024. Brain computer interface training with motor imagery and functional electrical stimulation for patients with severe upper limb paresis after stroke: a randomized controlled pilot trial. Journal of NeuroEngineering and Rehabilitation, 21(1), p.10.

These two useful references have been added (Introduction and Conclusions).

Further disscussuon of potential implications of this work in such scenarios should be further discussed.

DONE in the Conclusion section

Round 2

Reviewer 2 Report

Comments and Suggestions for Authors

Thank you for comments. 

Comments on the Quality of English Language

No

Author Response

no further requests were done for round 2 by rev 2

Reviewer 3 Report

Comments and Suggestions for Authors

Title: “The role of interoceptive sensitivity and hypnotizability in motor imagery”

Manuscript ID: brainsci-3107859-peer-review-v2

This work proposes a first analysis of the role of interoceptive sensitivity in the cortical activities associated with motor imagery. Concretely, the close relationship between interoceptive sensitivity and hypnotizability in motor imagery is explored, aiming to shed light on the embodied cognition concept. 

I appreciate the authors' answers to my feedback on the previous manuscript draft. In the following lines, I'll reiterate my earlier comments in black, highlight the authors' responses in blue, and then emphasize my feedback as "Reviewer" in red. 

A.      Abstract

1.       The manuscript's contribution ought to be presented in the present tense rather than the past. I am aware that the experiment has already been done. See lines 11–14 and 15–19 on page 1.

·       Authors: Done in the results part of the abs

o   Reviewer: The suggestion has been implemented by the authors.

2.       As the visual and kinesthetic modalities of the participants were classified according to the hypnotizability scores, it is expected to find in this section not only the literary but also the numerical results.

·       Authors: Groups hypnotizability scores have been included.

o   Reviewer: The suggestion has been incorporated into the manuscript. 

3.       Consider improving for this section the English writing style and spelling.

·       Authors: We checked the text.

o   Reviewer: The English writing style and spelling have been improved for this section.

B.      Keywords

4.       I suggest that the authors include MAIA among the keywords.

·       Authors: DONE

o   Reviewer: The suggestion has been incorporated into the manuscript. 

5.       It is more suitable to use "interoceptive sensitivity" instead of "interception."

·       Authors: DONE

o   Reviewer: The suggestion has been incorporated into the manuscript.

C.      Introduction

6.       Consider improving this sentence: "It is used in neuro-rehabilitation therapies, including brain-computer interface interventions; however, its efficacy is difficult to predict due to the large variability of motor imagery abilities in the general population." For example, “...interventions. However, its…”, or “...due to the large variability of motor imagery abilities across subjects.”

·       Authors: DONE

o   Reviewer: The suggestion has been implemented by the authors.

7.       Consider checking the beginning of this sentence: "Actual and effective imagined actions also share time duration..." Something is wrong. See on Page 1, line 39.

·       Authors: DONE

o   Reviewer: The suggestion has been incorporated into the manuscript.

8.       The introductory section should be rewritten, highlighting the contributions of this work, improving the English writing style and spelling, and logically organizing the sequence of paragraph contents. Reading through the introduction, I do not feel the cohesion between the ideas developed in each paragraph.

·       Authors: We have reorganized this section. In the present version of the manuscript, the logic map of the text is: 1)Motor imagery, 2) motor imagery and interoception, 3)hypnotizability and motor image, 4) hypnotizability and interoception.

o   Reviewer: The suggestion has been incorporated into the manuscript. 

9.       It would be interesting to add the related work section after the introductory one. This would allow the contributions of this article to be better highlighted.

·       Authors: Line 65-67: Nonetheless, no study has been performed on the role of interoceptive sensitivity in motor imagery, despite the well-recognized role of the insula, whose activity is associated with interoceptive accuracy and sensitivity  

The background of the study is presented in the Introduction: We have added, atits end,:  The expected results will highlight for the first time the role of  interoceptive sensitiv ity and hypnotizability  in motor imagery, which is a novel topic

o   Reviewer: I agree with the authors' point of view. The rewriting of the Introduction section already highlights the close works. 

10.   The keywords "Multisensory Assessment of Interoceptive Awareness (MAIA)" and EEG presented in the abstract were left without development or bibliography in the introduction.

Authors: Corrected. IS is mainly measured by the Multidimensional Assessment of Interoceptive Awareness (MAIA,  which consists of 8 scales (noticing, not distracting, not worrying, attention regulation, emotional awareness, self-regulation, body listening, trusting ) indicating the mode of the individual interpretation of interoceptive signals 

o   Reviewer: The suggestion has been incorporated into the manuscript.  

D.      Materials and Methods 

11.   According to what is stated on Page 2, lines 90–93, namely, "... with mediums representing 70% of the general population and highs and lows each representing 15% [32], how did the authors find the general population representation of 70%? 

·       Authors: The reference De Pascalis et al, 2000 indicates that this information has been obtained in a representative sample including 15% highs, 15 %lows, 70% mediums.

o   Reviewer: My doubts about this observation have been clarified.

12.   The results presented from line 93, Page 2, to line 96, Page 3, are not clear. In addition to the expected explanations, consider improving the English writing style and grammar.

·       Authors: The text has been updated and moved to the results section according to rev 2 comments

o   Reviewer: The suggestion has been incorporated into the manuscript.

13.   What is the relevance of this sentence beginning, "The experiments were conducted on the same day..." compared to the previous paragraph and the text that follows?

·       Authors: Authors are usually requested to indicate in which part of the day the experiments are performed owing to the several biological circadian cycles. At least 2 weeks after hypnotic assessment (to prevent expectancy effects), questionnaires administration and the experimental session were performed. The proneness to be deeply involved in cognitive tasks and interoceptive sensitivity were assessed through the Tellegen Absorption Scale (TAS, [33]) and the Multidimensional Assessment of Interoceptive Awareness (MAIA, [34]). the experiments were conducted between 9 an 12a.m. in a semi-dark and quiet room

o   Reviewer: My doubts about this observation have been clarified.

14.   According to what is said on Page 3, from lines 105 to 120, I can suspect that the conductor of the experiment had no software support to confirm that the imagined movements were actually executed. Was only the declaration of the subject, as said on lines 117–118, sufficient to approve the imagined task?  In similar experiments in the literature, these crucial moments of capture are monitored to observe clear differences between the cue signals and those of the baseline.

·       Authors: In the results section of the original version of the paper, we report the criteria which enabled us to consider the imagery tasks performed according to the literature. “...Based on both the functional equivalence model and on the literature showing lower desynchronization during imagery than during actual movement [38,39], we studied the conditions in which K and/or V power spectra were higher or non-significantly different from M, provided the presence of significant differences between B and V/K,

o   Reviewer: My doubts about this observation have been clarified.

15.   The caption of Figure 1 is very silent. Please, add more explanations about each diagram block.

·       Authors: Figure 1. Experimental design of the study. Blocks represent the experimental conditions, which were separated by a 30 sec rest. Each imagery condition was repeated 3 times.  

o   Reviewer: The suggestion has been incorporated into the manuscript.

16.   This section needs a thorough revision of English grammar and spelling.

·       Authors: DONE

o   Reviewer: The English writing style and spelling have been improved for this section.

17.   The text from Page 3, lines 125–132, and then lines 134–143 of Page 4 should be summarized for this section, and its entirety should be moved to the additional material. These instructions for the experiment subjects should be short but concise. Clear and brief.

·       Authors: Each imagery series was preceded by listening to the script describing the movement (Appendix A, Suppl El Mat). For Visual imagery the suggestions read:“…Now please imagine doing the same movement you did a few minutes ago. You can clearly see your left arm flexing up to the shoulder while your fingers touch your thumb one by one from the index to the little finger…”. For the kinesthetic imagery the suggestions read : “…Now, please imagine repeating the same movement you did a few minutes ago. You can feel the tension growing in your left biceps as it flexes up to the shoulder, while the muscles in your forearm start contracting and your fingers touch your thumb one by one from the index to the little finger …”.

o   Reviewer: The suggestion has been implemented by the authors.

18.   I am surprised to discover electrocardiogram (ECG) as a technique developed by the authors in the manuscript. Neither the abstract nor the introduction mention this technique.

·       Authors: Adding the evaluation of heart rate during cognitive tasks like motor imagery is suggested by Collet et al., 2011(cited in the paper introduction) in the light of the multidimensional assessment of motor imagery. Indeed, The efficacy of MI includes autonomic activation (heart rate, electrodermal activity), subjective reports, EEG signals and chronometric measures.

o   Reviewer: My doubts about this observation have been clarified.

19.   Some justifications for setting the sampling frequency at 500 Hz and all impedances below 30 kΩ?

·       Authors: This is the best setting offered by our recording system. High sampling frequency is often used in literature to improve the temporal resolution of high frequency bands. In addition, the signals will be used for further analysis with different aims (Cohen M, MIT Press, Cambridge, Massachussetts, 2014)

o   Reviewer: My doubts about this observation have been clarified.

20.   Section 2.4, related to signal acquisition and analysis, should be rewritten by presenting step-by-step each technique utilized in the chain processing. See on Page 4.

·       Authors: This is indeed presented in the text.

There is already evidence to question the authors' results because of this lack of clarity. In particular, why was the two-way least-squares FIR filtering with a bandwidth of 0.5–45 Hz selected? Although alpha and beta (7–30 Hz) are the bands of interest that are specifically mentioned in the summary section. 

·       Authors: We filtered just to avoid high frequency muscle artifacts and low frequency thermoregulatory, respiratory and other electrical artefacts and to maintain lower and higher frequncies for further analyses

Why did the authors use the spline interpolation technique to substitute noisy signals from specific channels?

·       Authors: According to   Perrin et al., 1989, this method is well fit to mantain the spectral characteristic of signals.

The findings to be acquired are tainted by the gathering of additional data since the spline technique—even cubic—has limitations when it comes to extrapolation. The signals were downsampled to 256 Hz; why?

·       Authors: In the present study this downsampling satisfied our aims and reduced required computing power.

With what settings was the ICA algorithm applied?

·       Authors: We used Extended ICA algorhythm and selected 20 components explaining the highest amount of variance.

o   Reviewer: I recommend to the authors including this information in the manuscript.

How many independent components were employed for the analysis for each channel and each subject, and why was that decision made?

·       Authors: According to Delorme and Makeig (cited in the text) the optimal number of independent components is between half  and three quarters of the electrodes number in order to identify  artifacts without reducing EEG signal information

o   Reviewer: I recommend to the authors including this information in the manuscript.

 Why is there another downsampling to 128 Hz?

·       Authors: This part of the analysis was automatically performed using an open-source MATLAB toolbox EEG-Beats [37], which was applied only to the ECG signal

o   Reviewer: My doubts about this observation have been clarified.

21.   Section 2.6 is expected to present statistical analysis, but it is empty. Please consider including statistical variables, as claimed by the section title.

·       Authors: The statistics paragraph present in the original version of the paper, following the section 2.5.2. Paragraph 2.5 contais the studied variables before the experimental session SHSS, MAIA, TAS) and during the experimental session (EEG alpha, low beta, high beta, subjective reports, chronometric indices, heart rate.

o   Reviewer: My doubts about this observation have been clarified.

E.       Results

In addition to the doubts, I expressed regarding the findings of the research in observation 10 of the Materials and Methods section, I completely disagree with any findings made in this section for the reasons listed below:

·       Authors: We have clarified all the reviewer’s doubts regarding the following points and already listed in the methods section

o   Reviewer: I agree with the authors’ reply.

22.   The procedure for distinguishing between alpha and beta waves is not rendered clear in the manuscript (refer to Tables 2 and 3, pages 7 and 8).

·       Authors: Two separate tables are presented for alpha and low/high beta ANOVA results. They are indicated in the text. We have updated the tables to improve the clarity

o   Reviewer: My doubts about this observation have been clarified.

23.   Data used was a combination of data generated by the spline interpolation and those really recorded.

·       Authors: We have clarified all the reviewer’s doubts regarding this point and already listed in the methods section. Moreover, we have checked that none of the interpolated channels was used in the present study (we only used the electrodes F3, F4, C3, C4, P3, P4, PO3, PO4)

o   Reviewer: My doubts about this observation have been clarified.

24.   The PSD algorithm's spectral analysis lacks enough justification and explanation.

·       Authors: This is written in the Methods section. However, we wish to clarify that we used Hamming windowing rather than a rectangular window to avoid spectral distortions due to borders discontinuity, with 4 sec epochs as a compromise between temporal and frequency resolution to obtain an optimal resolution for alpha and beta bands , 50% overlap owing to the necessity to reduce the spectral variability.

o   Reviewer: Consider adding this information in the manuscript.

25.   The number of subjects is not disclosed in the results that are presented. It's unclear if each experiment subject completed the task accurately.

·       Authors: All participants completed the study. Reported in the text

o   Reviewer: My doubts about this observation have been clarified. 

26.   Do the presented averages relate to the total number of sessions, subjects, and number of channels?

·       Authors: Yes, they do ( 1 session was performed. PSD was studied at F3, F4, C3, C4, P34, P4, PO3, PO4 for each condition.)

“PSD was averaged across the three visual and three kinesthetic condition and across the two actual movement condition. Signals from frontal and central, as well as parietal and occipital electrodes were averaged for every to obtain two regions (fronto-central, FC, and parieto-occipital,PO ) for every condition”

o   Reviewer: My doubts about this observation have been clarified.

27.   of the 58 references provided in the manuscript, 5 are incomplete (missing number of pages, confusion of the year of publication, etc.), 3 cannot be found on the publications’ hub, and 16 are old more than 12 years.

·       Authors: We corrected the mistakes. Pisa lab, involved in studies of the physiological correlates of hypnotizability, is the only lab in the world addressing this topic. Thus, it is obvious that we built our research based on our earlier research.

o   Reviewer: The suggestion has been implemented by the authors.

28.   The quality of the writing in English is below average. The authors should revise the entire manuscript.

·       Authors: DONE

o   Reviewer: The English writing style and spelling have been improved for this section.

29.   13 Self-citations by authors of works.

·       Authors: All these referencences are required.

o   Reviewer: I find it exaggerated that the authors use 15 self-references, while emphasizing that the citation of themselves affects the value of their work. Since they cite works, they themselves have already published, I would advise them to cite only one of their references rather than two at once.

F.       Discussion

30.   It is missing a discussion about the ECG-developed technique.

·       Authors: We cannot discuss this point because we just applied already developed algorithms to simply identify heart rate.

o   Reviewer: I disagree with the authors’ point of view. That is, the authors used a technique whose significance or contribution was not evaluated in the development of their method. In other words, with or without the ECG mixture, would the achieved results be the same (ablation study)? To guide the authors, I suggest including in the manuscript the authors' response to observation 18.

31.   Unless I am mistaken, I did not find in the manuscript a comparison of the results presented by the authors with those published in the related literature, nor a rational discussion resulting from them.

·       Authors: No earlier findings are present in the specific literature. As we mentioned in the text, this study is a novel approach to the relationship between hypnotizability, motor imagery, and interoceptive sensitivity. A more detailed discussion of EEG findings would have confused the aim of the study and was useless to our aim

o   Reviewer: I understand the authors' argument. But then, my insistence on reducing the number of self-citations makes perfect sense. I also understand that the present study is an innovation because, already in observation number 9, the authors have provided a justification in this sense. 

G.      Limitations and conclusions

32.   Avoid putting references in the conclusion because this section summarizes your work by essentially recalling the goals of your work, the method developed, the results obtained, the limitations, and future projections of your work.

·       Authors:  DONE

o   Reviewer: The suggestion has been implemented by the authors.

33.   The limitations presented on page 11, lines 360–363, sufficiently show how the results presented are not robust and consistent. My advice would be to encourage authors to return to deep and methodical testing.

·       Authors: We think to have replied to and clarified each point addressed by this reviewer, and to have matched the rev 1 and rev 2 suggestions.

o   Reviewer: The suggestion has been implemented by the authors.

In summary: For the next round of revision, I would like to have feedback from the authors regarding my new suggestions for observations 20, 24, 29, and 30.

Recommendation: Minor revisions.

Author Response

Comment 20:Section 2.4, related to signal acquisition and analysis, should be rewritten by presenting step-by-step each technique utilized in the chain processing. See on Page 4.

  • Authors: This is indeed presented in the text.

There is already evidence to question the authors' results because of this lack of clarity. In particular, why was the two-way least-squares FIR filtering with a bandwidth of 0.5–45 Hz selected? Although alpha and beta (7–30 Hz) are the bands of interest that are specifically mentioned in the summary section. 

  • Authors: We filtered just to avoid high frequency muscle artifacts and low frequency thermoregulatory, respiratory and other electrical artefacts and to maintain lower and higher frequencies for further analyses

Why did the authors use the spline interpolation technique to substitute noisy signals from specific channels?

  • Authors: According to Perrin et al., 1989, this method is well fit to maintain the spectral characteristic of signals.

The findings to be acquired are tainted by the gathering of additional data since the spline technique—even cubic—has limitations when it comes to extrapolation. The signals were downsampled to 256 Hz; why?

  • Authors: In the present study this downsampling satisfied our aims and reduced required computing power.

With what settings was the ICA algorithm applied?

  • Authors: We used Extended ICA algorhythm and selected 20 components explaining the highest amount of variance.
  • Reviewer: I recommend to the authors including this information in the manuscript.

DONE

How many independent components were employed for the analysis for each channel and each subject, and why was that decision made?

  • Authors: According to Delorme and Makeig (cited in the text) the optimal number of independent components is between half  and three quarters of the electrodes number in order to identify  artifacts without reducing EEG signal information

o   Reviewer: I recommend to the authors including this information in the manuscript.

 DONE

In detail:

Suggestions related to points 20 and 24 have been addressed in “Methods” section, paragraph 2.4, lines 169-178: “Individual channels showing quality decline (due to instability or loss of contact with the scalp during recordings) were visually identified and replaced with signals obtained via spline interpolation, a method generally implemented to maintain the spectral characteristics of the signal [45]. To remove residual artifacts, values exceeding the range of -70–70 mV were discarded. Retained EEG signals were downsampled to 256 Hz to reduce the required computing power.̲ Signal was then submitted to Independent Component Analysis (ICA, [46]) to remove ocular, heart, and muscular artifacts in each subject. We used Extended ICA algorithm and selected 20 components explaining the highest variance, since the optimal number of components should be between half and three quarters of the electrodes number, thus balancing between identifying sufficient variance and avoiding overfitting”.

Comment 24:The PSD algorithm's spectral analysis lacks enough justification and explanation.

  • Authors: This is written in the Methods section. However, we wish to clarify that we used Hamming windowing rather than a rectangular window to avoid spectral distortions due to borders discontinuity, with 4 sec epochs as a compromise between temporal and frequency resolution to obtain an optimal resolution for alpha and beta bands, 50% overlap owing to the necessity to reduce the spectral variability.
  • Reviewer: Consider adding this information in the manuscript.

Comment 29: Comments related to point 29 have been addressed in “Methods” section, paragraph 2.4, lines 182-185: “PSD estimation was performed with a Hamming window of 4-second length, as a compromise between frequency and temporal resolution, and 50% overlap to reduce spectral variability. The choice of using Hamming window rather than rectangular window was justified by the necessity to avoid spectral distorsions due to border discontinuity”. 13 Self-citations by authors of works.

  • Comment 30o   Reviewer: I find it exaggerated that the authors use 15 self-references, while emphasizing that the citation of themselves affects the value of their work. Since they cite works, they themselves have already published, I would advise them to cite only one of their references rather than two at once.

 We have erased all the references related to information contained in a book chapter (Santarcangelo, 2004), We removed:

- Presciuttini, S.; Curcio, M.; Sciarrino, R.; Scatena, F.; Jensen, M. P.; Santarcangelo, E. L. Polymorphism of Opioid Receptors Μ1 in Highly Hypnotizable Subjects. International Journal of Clinical and Experimental Hypnosis 2018, 66 (1), 106–118. https://doi.org/10.1080/00207144.2018.1396128;

- Spina, V.; Chisari, C.; Santarcangelo, E. L. High Motor Cortex Excitability in Highly Hypnotizable Individuals: A Favourable Factor for Neuroplasticity? Neuroscience 2020, 430, 125–130. https://doi.org/10.1016/j.neuroscience.2020.01.042;

- Ibáñez-Marcelo, E.; Campioni, L.; Phinyomark, A.; Petri, G.; Santarcangelo, E. L. Topology Highlights Mesoscopic Functional Equivalence between Imagery and Perception: The Case of Hypnotizability. NeuroImage 2019, 200, 437–449. https://doi.org/10.1016/j.neuroimage.2019.06.044.

Point 30 has been addressed in lines 312-315 of the “Discussion” section: “The multidimensionality of motor imagery assessment is justified by the huge variety of its psychophysiological correlates, such as autonomic activation (heart rate, electrodermal activity), subjective reports, EEG signals and chronometric measures”.

Reviewer 4 Report

Comments and Suggestions for Authors

I understand that the nature of the task prevents the display of classical ERD(S) plots however some sort of visualization of the obtained EEG patterns is necessary for electrophysiological validation of the results and ERD(S) patterns obtained.

I suggest the following:

Display the topographic maps of alpha and beta PSD averaged for each condition and group. 

These plots should display the expected lateralization according to the task being executed. 

Moreover, such plots should further confirm the regions of interest that the authors chose to analyze.

Author Response

I understand that the nature of the task prevents the display of classical ERD(S) plots however some sort of visualization of the obtained EEG patterns is necessary for electrophysiological validation of the results and ERD(S) patterns obtained.I suggest the following:

Display the topographic maps of alpha and beta PSD averaged for each condition and group. 

These plots should display the expected lateralization according to the task being executed. 

Moreover, such plots should further confirm the regions of interest that the authors chose to analyze.

We have produced topoplots.  We wish to underline that, according to current evidence, side effects ca be scarcely lateralized during imagery tasks

Round 3

Reviewer 4 Report

Comments and Suggestions for Authors

Authors have addressed all my comments